# Dust-producing weather patterns of the North American Great Plains

Stuart Evans[1]

[1]University at Buffalo

**Correspondence:** Stuart Evans (stuartev@buffalo.edu)

**Abstract.** The North American Great Plains are a semi-arid and windy environment prone to dust events that produce a variety of hazards to public health, transportation, and land degradation. Dust has substantial spatial variability across the plains, and the weather responsible for that dust is understudied in most of the plains, especially the North and East. Here we identify specific weather patterns associated with dust occurrence across the plains. We make use of an atmospheric classification that defines 21 weather patterns for the Great Plains that includes various stages of warm and cold frontal passages, northerlies, anticyclones, and summertime patterns not associated with mid-latitude cyclones. We use the time series of weather pattern to composite satellite and station daily dust observations from 2012-2021. We calculate average dust occurrence for each weather pattern, the contribution of each pattern to local dust loads, and identify the specific weather patterns most important to each location and subregion. We find no single weather pattern is responsible for dust occurrence in the plains, but that different patterns are responsible for dust in different subregions of the Great Plains. Passing cold fronts are most responsible for dust events in western Texas and New Mexico, southerlies are responsible in the northeastern plains of Iowa and the Dakotas, and summer weather patterns produce the majority of dust in the High Plains from Colorado to Canada. Identifying the dust-producing weather patterns of particular subregions is a valuable step toward understanding dust variability and improving dust predictions, both present and future.

 # 1   Introduction

The North American Great Plains has a long and varied history as a dust source. During the mid-Holocene, North America experienced mega-droughts that lasted for decades and made the Great Plains into strong dust sources (Cook et al. (2007); Cook et al. (2016)). The legacy of this dusty period can still be found in the Sand Hills of western Nebraska where layers of wind-blown sediment from the mid-Holocene are only thinly covered by soil and vegetation today (Miao et al., 2007). Famously, during the Dust Bowl event of the 1930s the southern plains were transformed into an intense dust source by the combined effects of drought and vegetation loss from farming practices not suitable for the region (Schubert et al., 2004; Cook et al., 2009). Today the Great Plains as a dust source are a complex system of both natural and anthropogenic forces (Chen et al., 2018; Ginoux et al., 2012). Conservation tillage, groundwater irrigation, and soil conservation districts have prevented the region from experiencing subsequent dust bowls despite periods of drought (Basara et al., 2013; Angadi et al., 2016; Hansen and Libecap, 2004), but intensive agricultural development has nonetheless enhanced anthropogenic dust emission in the region (Lambert et al., 2020; Kandakji et al., 2021). The climate in most of the region is semi-arid and subject to strong winds, so natural dust emission remains an important part of the regional dust cycle. This climate predisposes the region to act as a dust source and climate variability modulates the strength of that source, but the immediate cause of dust emission is individual weather events in the region (Aryal and Evans, 2022; Pu and Ginoux, 2018, 2017; Achakulwisut et al., 2017).

Previous research on dust variability in the Great Plains has primarily focused on climate and climate variability of the region. Seasonally, dust in all parts of the Great Plains is at a minimum in winter, has a spring peak in the southern High Plains of Texas, New Mexico, and Colorado, and a summer peak for the plains east and north of the Texas Panhandle (Hand et al., 2017; Aryal and Evans, 2022). On interannual timescales, El Niño, the Pacific Decadal Oscillation, and Pacific-North America pattern have all been identified as contributing to spring dust variability via their impacts on rainfall patterns (Achakulwisut et al., 2017). This is broadly in agreement with findings by studies that have investigated the relationships between dust occurrence and seasonal precipitation, wind speed, drought, and vegetation (Aryal and Evans, 2022; Pu and Ginoux, 2017, 2018; Arcusa et al., 2020)).

In contrast, there is relatively limited research on the specific weather patterns that are the proximate cause of dust emission and transport in the plains. Where there has been such research, it has led to the identification of dust weather for specific subregions of the United States. For example, "Albuquerque Lows", wherein a cold front associated with an upper-level trough and a surface low in Colorado sweeps across New Mexico and the Chihuahuan desert, have been identified as the primary cause of dust events in El Paso and the Southern High Plains (Novlan et al., 2007; Rivera et al., 2009). Pu and Ginoux (2018) showed that summertime dusty days in the central Great Plains (north Texas through Kansas) are associated with a westward extension of the subtropical high and intensification of the low-level jet. Outside of the Great Plains, dust events in the Great Basin of Utah are primarily caused by passing troughs with surface lows along the Nevada-Idaho border (Hahnenberger and Nicoll, 2012), and dust events in Arizona are most commonly caused by either frontal passages or thunderstorms, depending on which part of the state (Brazel and Nickling, 1986). These works are invaluable in understanding the origins of dust in

particular areas, but there remain many understudied regions. In this study we aim to comprehensively identify such patterns for all parts of the Great Plains in all seasons.

The importance of recognizing dust weather across the Great Plains is underscored by the wide variety of human impacts from dust in the region, especially regarding respiratory health and travel hazards. Dust events in El Paso, Texas are associated with increased hospitalizations for asthma and bronchitis (Grineski et al., 2011), and worldwide, exposure to mineral dust increases the risk of cardiovascular disorders and lung cancer (Goudie, 2014; Giannadaki et al., 2014). Dust originating in the Southwest US has also been shown to be associated with the fungal spores that transmit valley fever (Tong et al., 2017). Many

dust sources in the region are near highways (Li et al., 2018), frequently affecting travel in the region by restricting visibility, and leading to highway closures, traffic accidents, and approximately 21 deaths per year (Tong et al., 2023). Many of these impacts, especially travel hazards and acute respiratory illness, are short-lived in time and only occur during and immediately following dust events. Again, the timescale of these impacts underlines the importance of understanding dust at the timescale of weather events in addition to seasonal and climatic timescales.

In this manuscript we identify the specific weather patterns that are responsible for dust occurrence in different portions of the Great Plains. We do this by comparing a time series of weather patterns (Evans et al., 2017) to both satellite and station-observed time series of dust occurrence and identifying the patterns which produce the most dust and those which produce the largest percentage of a region's dust. We describe the classification of the weather patterns, the patterns themselves, and the dust observations in Section 2, and the results of comparing those time series in Section 3. Section 4 summarizes our findings

and discusses additional implications.

## 2    Methods

### 2.1    Classification Process

The weather patterns used in this study were originally defined in Evans et al. (2017), hereafter E17, which contains full details of the classification process and the results. We briefly summarize here the key details of the classification process, and the

results of that process, i.e. the weather patterns themselves. E17 defined weather patterns in the Great Plains region for the purpose of understanding cloud and radiation properties at the Atmospheric Radiation Measurement (ARM) Program's Southern Great Plains observation site in Oklahoma (Muhlbauer et al., 2014; Zhao et al., 2017). E17 used a previously-developed iterative clustering algorithm (Evans et al., 2012; Marchand et al., 2009, 2006) applied to three-dimensional ERA-Interim reanalysis fields for a region spanning south Texas to northern Nebraska and eastern Colorado to eastern Missouri (29.25 – 42.75

75    °N, 90.75 – 104.25 °W). The fields to represent the weather of the region were air temperature, relative humidity, the u- and v-components of wind, and surface pressure. The fields were sampled on a 9x9 grid spanning 13.5° of latitude and longitude and on seven pressure levels spread through the troposphere. This three-dimensional description of the region's weather was sampled four times daily from 1996-2010, producing 19,476 snapshots of the state of the atmosphere for classification. A k-medians classification algorithm was used to identify and define commonly occurring weather patterns for the region. The

patterns were tested for within-pattern consistency and inter-pattern distinctness using independent cloud radar data from the

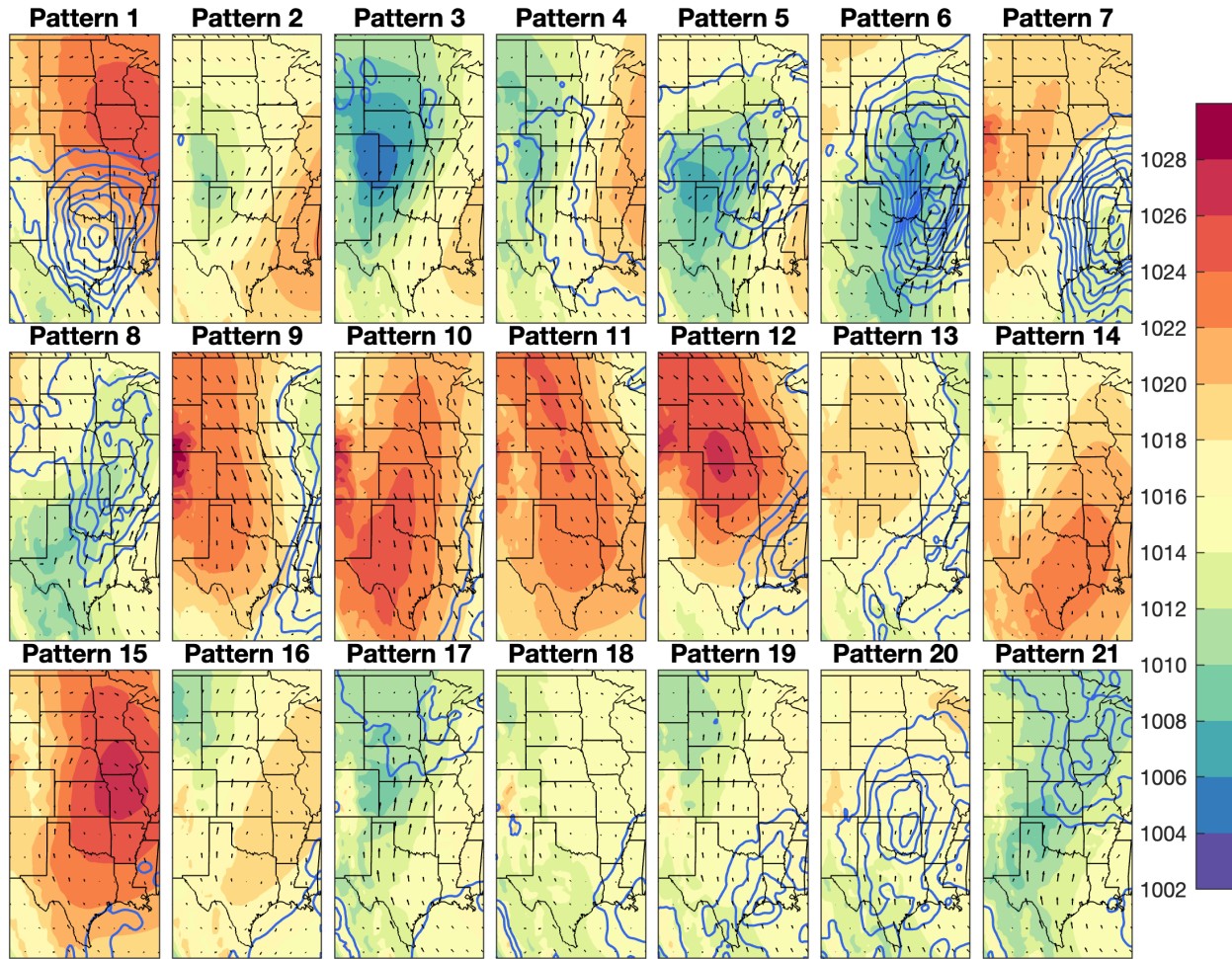

**Figure 1.** Composite 2012-2021 ERA5 properties for each of the 21 weather patterns. Underlying color shows sea level pressure (mb), blue contours are precipitation in 2 mm/day increments beginning at 4 mm/day, and arrows show 875mb wind speed and direction, with longest arrows representing 17.5 m/s.

ARM site. This process was iterated upon in an automated manner, varying the number of patterns, until a final set of weather patterns had been defined, each of which passed the statistical tests for consistency and distinctiveness.

The E17 classification process produced a set of 21 weather patterns for the region. Figure 1 shows a selection of composite meteorological values – sea level pressure, 875mb flow, and precipitation - for each weather pattern. Additional values that help to fully describe the weather patterns (500mb flow, 2m temperature) are shown in Figure A1. These patterns could be placed in five broad categories describing the weather in the region, based primarily on the low-level flow in the southern plains: southerlies and warm sectors (Patterns 1-4), cold fronts (Patterns 5-8), post-frontal northerlies (Patterns 9-13), high-pressure systems (Patterns 14-16), and summer weather (Patterns 17-21). Some patterns are borderline, e.g. Pattern 9 has the trailing

end of a cold front on the eastern boundary of the domain, but is placed in the northerlies as that is the flow pattern covering most of the region. The first four categories of weather pattern represent phases of passing synoptic-scale weather systems that predominate Great Plains weather outside of the summer months and have a predictability to them. Southerly or warm sector patterns (Patterns 1-4) are followed by cold front patterns (5-8) as the associated low pressure system and upper-level trough travel from west to east across the region. Cold northerly patterns (9-13) and high-pressure anticyclones (14-16) then follow as an upper-level ridge passes. The low-level flow and precipitation shown in Figure 1 help to identify the subregions of the Plains most likely to experience strong winds and dry conditions. Figure 1 also shows that within a category the differences between patterns are typically a matter of geographic shifts of the feature, e.g. how far north the southerlies extend (Patterns 2 and 3), cold fronts further east or west (Patterns 5, 6, and 7), or high pressure systems that are shifted north or south (Patterns 14 and 16).

The patterns for E17 were classified for the period from 1996-2010. As the patterns are defined by reanalysis, the time series of pattern can be readily extended in time. In order to bring the time series up to the present, the original patterns were matched to ERA5 reanalysis. This allows the categorization of the weather pattern for the entire ERA5 period. In this study the time period analyzed is the ten years from 2012-2021. This period is chosen to match the satellite dust observations used (next section) that begin in 2012. As the dust observations have daily resolution, we classify each day as belonging to only the weather pattern occurring closest in time to the satellite observation time, i.e. each day is assigned to the pattern identified at local noon.

## 2.2  Dust Observations

### 2.2.1  Satellite Data

We use the Visible Infrared Imaging Radiometer Suite (VIIRS) Deep Blue daily 1°x1° aerosol product (Hsu et al., 2019; Sayer et al., 2019) to provide dust observations each day for the years 2012-2021 for the region. VIIRS aerosol data compares with AERONET data as well as MODIS data (Hsu et al., 2019; Sayer et al., 2019) and comes with an additional data product of aerosol type. Retrievals are classified as containing dust if they are not classified as smoke (based on reflectivity at multiple wavelengths and brightness temperature) and if their Angström Exponent is less than 0.5, indicating the presence of coarse mode particles. As such, a dust-classified retrieval indicates dust particles were the predominant aerosol in the atmospheric column. We represent dust occurrence in the Great Plains with the number of retrievals within each gridbox classified as dust. Limiting ourselves to only these retrievals undercounts the occurrence of dust, as there are days with mixed aerosol species, but also provides confidence that the data being composited by weather pattern are not other aerosols. The VIIRS instrument orbits aboard the Suomi-NPP satellite, which has an overpass time of 1:30 PM local time. As such our analysis is of dust events that initiate in the morning or midday, or of long-lasting dust events. Short-lived dust events that initiate after the overpass, or that occur beneath clouds, are not captured in this data. The identification of the retrieval as dust is also a column value, and thus does not indicate the altitude of the dust particles or whether the location of observation is also the location of origin.

We discuss the impact of these limitations on our results in Section 4. Nonetheless, these data remain a valuable source of information on the occurrence of dust in the Western US, particularly through their complete spatial coverage.

Each day for the period of study is classified as one of the 21 weather patterns, allowing the VIIRS data to be composited according to weather pattern. This produces both spatial distributions of dust occurrence for each weather pattern and temporal distributions of weather pattern for the occurrence of dust in any particular location. Ten years of daily classification yields 3,653 days of dust observations that are composited by weather pattern, producing robust statistics for the patterns.

### 2.2.2 Station Data

We complement the VIIRS satellite data with station data from the Interagency Monitoring of Protected Visual Environments (IMPROVE) program (Malm et al., 1994; Hand et al., 2011). IMPROVE measures surface-level particulate concentrations and elemental composition on federal lands across the US. In this case we use the 29 stations between the Rocky Mountains and 90 °W longitude that were operating during the period of study. IMPROVE stations use an air pump to bring air through an inlet and filters of different sizes to measure particulate mass per volume of air. The pump operates for 24 hours at a time once every three days, providing a 24-hour integrated measure of surface particulate concentration. Using an empirical formula based on the measured elemental composition of the particulate matter, IMPROVE provides a measure of fine soil concentration ($\mu g/m^3$, (Malm et al., 1994; Hand et al., 2019)) which we use as a measure of dust in this study. Once again, we composite these data according to the weather pattern at local noon. While these data have their own limitations, they help control for the observation gaps of satellite data as they are not susceptible to clouds or overpass time. Further, as they measure surface level concentration, IMPROVE data are more directly related to the human impacts of dust.

## 3 Results

### 3.1 Mean dust occurrence

Figure 2 shows the mean daily retrievals identified as dust by VIIRS for each of the 21 weather patterns, e.g. the average Pattern 5 day has 10 retrievals marked as dust in far west Texas. Taken collectively, they show dust in the US occurs most frequently over the western Great Plains in the lee of the Rockies, in agreement with previous findings from MODIS (Ginoux et al., 2012). Taken individually, the weather patterns show substantial variety in the spatial distribution and frequency of dust occurrence. Many patterns, such as Patterns 8 and 12, show dust as very rare across the entire region, perhaps not surprising as the central US is not always a dusty region. Some patterns, however, are strongly connected with dust in particular locations. We focus on a selection of regions with strong connections between dust occurrence and weather pattern in the following subsections.

Figure 3 shows the mean fine soil concentration measured by the 29 IMPROVE stations for each weather pattern. Direct comparison to VIIRS results requires caution as they are two different measures of dustiness; they agree nevertheless in many regards with the VIIRS results, but notably vary in others. Taken together, the stations show the western and especially the southern plains as the dustiest parts, again in agreement with prior research (Hand et al., 2017). Once again, many weather

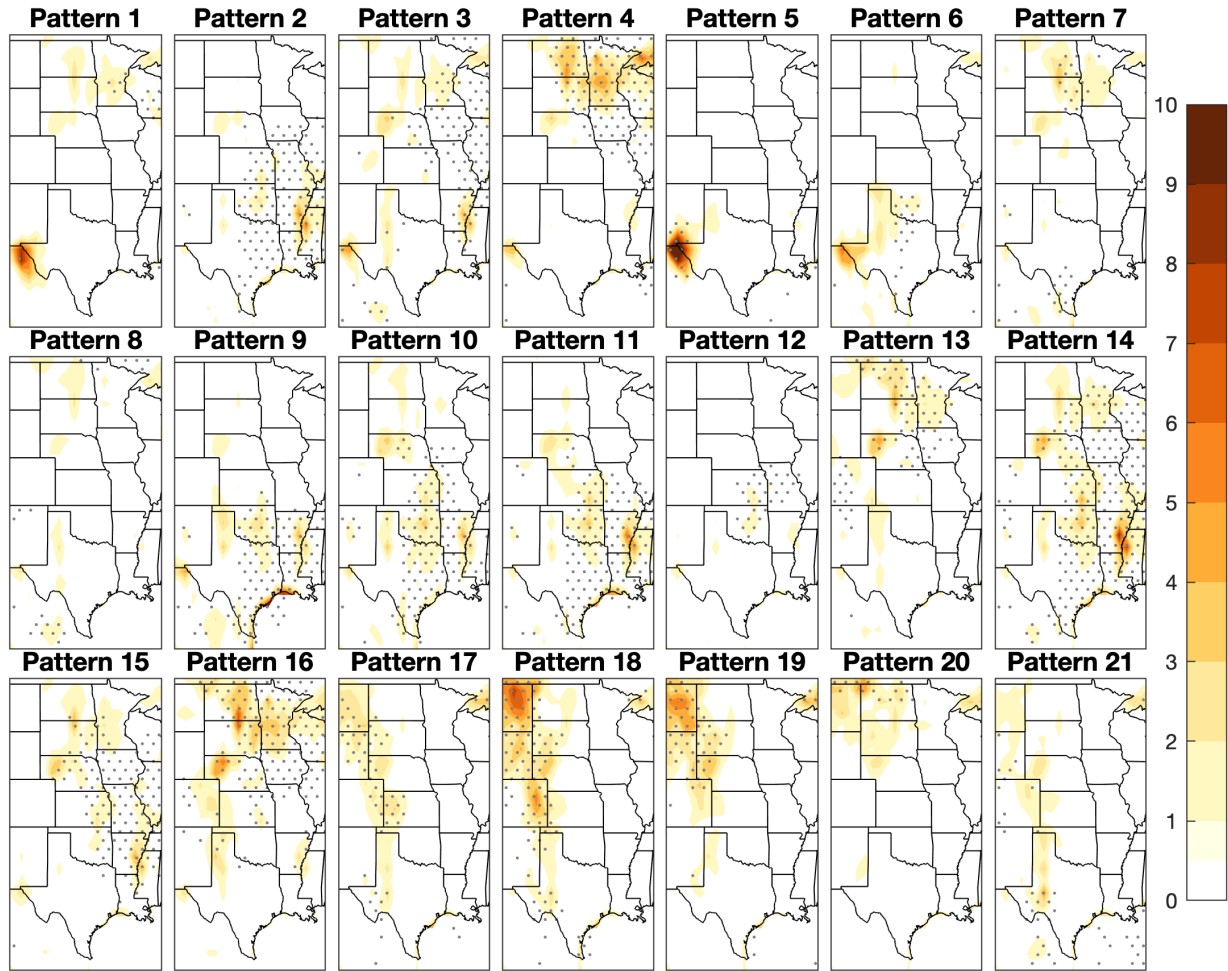

**Figure 2.** Average number of daily VIIRS retrievals within each 1°x1° gridcell identified as dust for each weather pattern. Each panel represents the average of all days identified as each pattern (100 to 322 days, average of 171) from 2012-2021. Stippling indicates regions where the pattern average exceeds the average of the full dataset with 95% confidence, determined by a one-tailed t-test.

patterns show very little dust across most of the plains, while others have specific regional signals. Compared to Figure 2 two underlying differences stand out. First, the southern plains, and especially the southwestern plains of West Texas and New Mexico, show consistently high values of dustiness that are not seen in the dust frequency results from VIIRS. Second, the northern plains from Montana to Minnesota vary more cohesively in the IMPROVE results while the VIIRS results often show only the northwestern or northeastern plains with dust. We discuss these differences further within the context of the different types of observation in Section 4. We discuss individual patterns and their importance to particular regions below.

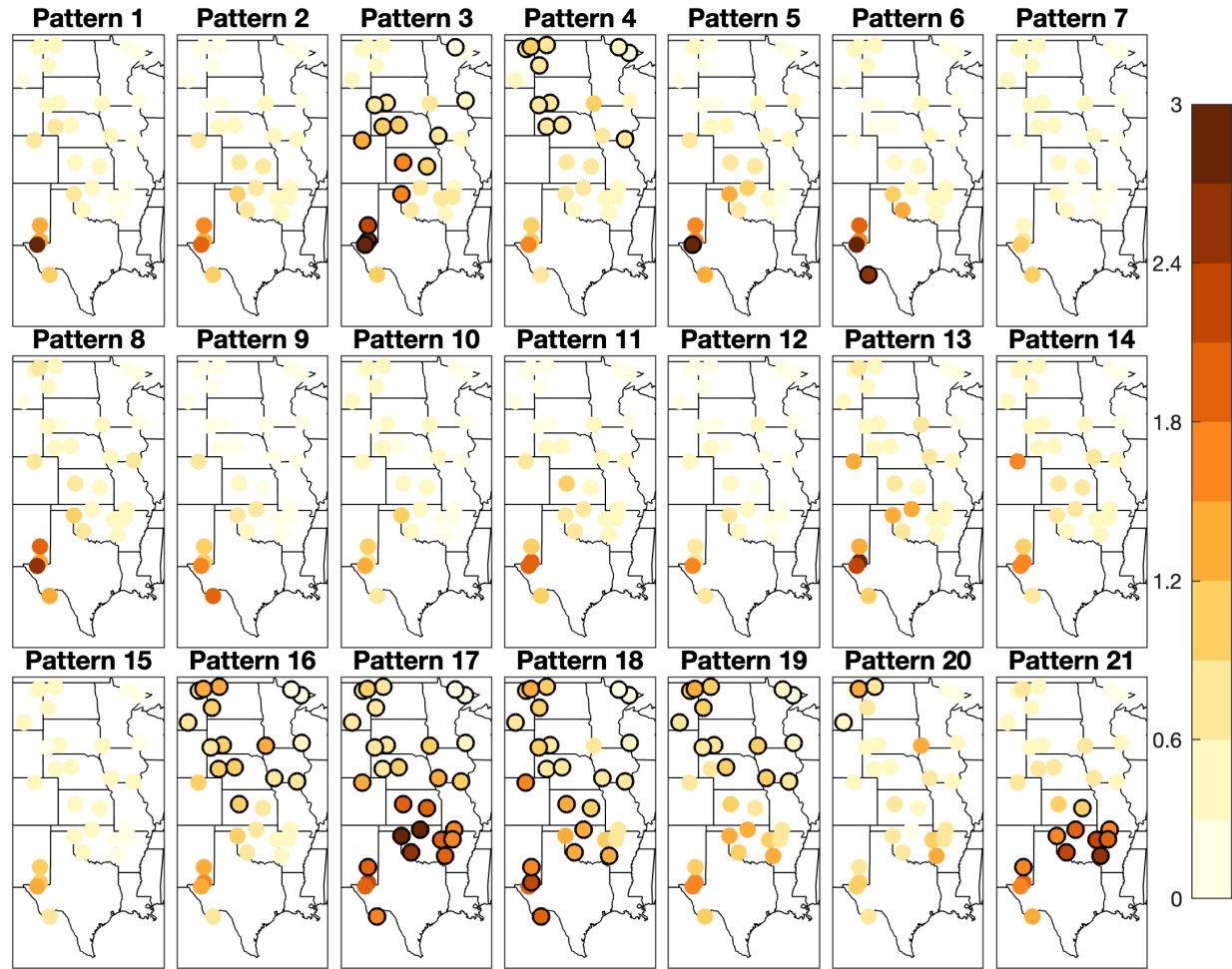

**Figure 3.** Average fine soil concentration ($\mu g/m^3$) at IMPROVE stations for each weather pattern. Each panel represents the average of all days identified as each pattern from 2012-2021. Black outlined points indicate stations where the pattern average exceeds the average of the full dataset with 95% confidence, determined by a one-tailed t-test.

### 3.1.1 West Texas

Most notable in the VIIRS data is Pattern 5, which produces an intense bullseye of dust over the city of El Paso and the surrounding area. This feature is also present in IMPROVE data, with the Guadalupe Mountains station (far west Texas) showing significantly elevated dust concentrations. Pattern 5 represents a cold front over the Texas Panhandle leading a deep upper-level trough over the Rocky Mountains (Evans et al., 2017). This produces strong southwesterly winds over northern Mexico that bring intense dust plumes from the Chihuahuan Desert across West Texas and southern New Mexico. This pattern and resulting dust storm is very similar to the "Albuquerque Low" weather system identified by Novlan et al. (2007) as a key

contributor to dustiness in the El Paso region. VIIRS also finds this pattern to be the primary contributor to dust occurrence in the region. Figure 4 shows the percentage of all dust retrievals that occur during each weather pattern. Indeed, 30-50% of all VIIRS retrievals classified as dust in the El Paso region occur during Pattern 5. IMPROVE, however, only shows Pattern 5 as being responsible for a modest fraction of the dusty days in the region, defined as days when the fine soil concentration exceeds 1 $\mu g/m^3$. This difference is primarily due to IMPROVE finding the West Texas and New Mexico stations to be dusty during most weather patterns, thus diluting the importance of any single pattern. Pattern 6 also shows frequent dust over West Texas and the Texas Panhandle in both datasets. This pattern frequently follows Pattern 5 in time, with the same cold front and upper-level trough as in Pattern 5 having shifted eastward as the synoptic weather event evolves. Winds remain strong over West Texas, producing additional uplift of dust around El Paso, and dust that was previously uplifted during Pattern 5 has been advected to the north and east, across West Texas and into the Panhandle and Oklahoma. This can also be seen in Figure 4 as a major contributor of dust occurrence in the Llano Estacado of eastern New Mexico and the Texas Panhandle, as well as southwestern Oklahoma.

### 3.1.2   Oklahoma and the southeastern plains

The region with the largest divergence between VIIRS and IMPROVE data is the southern plains of Oklahoma and Kansas. VIIRS rarely identifies dust in this region and when it does find dust to be present it does so across a variety of types of weather, including southerlies (Pattern 2), northerlies (Patterns 10 and 11), and anticyclones (Pattern 14). In contrast, IMPROVE shows both high dust concentrations and frequent dusty days during the summer weather patterns, particularly Patterns 17, 18, and 21. A likely explanation for the lack of summer dust detections in the VIIRS data is the time of satellite observation. Late afternoon convection that can drive dust uplift during summer is not observed by VIIRS with its 1:30pm local overpass time. The 24-hour collection time of IMPROVE catches these summer dust events, providing the values seen in Figure 3. Additionally, seeing these summer events raises the background dust level such that the dust contributed by the patterns VIIRS identifies is no longer significantly elevated (Figure 3) or and important contribution of dust (Figure 4).

### 3.1.3   Minnesota and the eastern Dakotas

While not as dusty as the southern plains, the northeastern plains have a more varied range of weather patterns that produce dust in the region. VIIRS and IMPROVE are largely in agreement in this region. Pattern 4 is the most important contributor of dust in the region in both datasets (Figure 4), showing dust over Minnesota and the eastern Dakotas, as well as smaller amounts of dust in surrounding areas of Iowa, Wisconsin, and Canada. This pattern features strong southerlies in advance of a surface low (Fig. 1) that has brought warm temperatures to the northern plains (Fig. S1). While the surface winds are stronger in the southern plains than in the north, the southerlies carry with them moisture from the Gulf of Mexico that brings precipitation that suppresses dust emission further south. Minnesota and the Dakotas are north of the advected moisture, where they still experience enhanced surface winds, but not the precipitation associated with it. Pattern 3 has similar meteorology that also produces dust in the region, but somewhat weaker winds makes Pattern 3 a weaker dust contributor than Pattern 4.

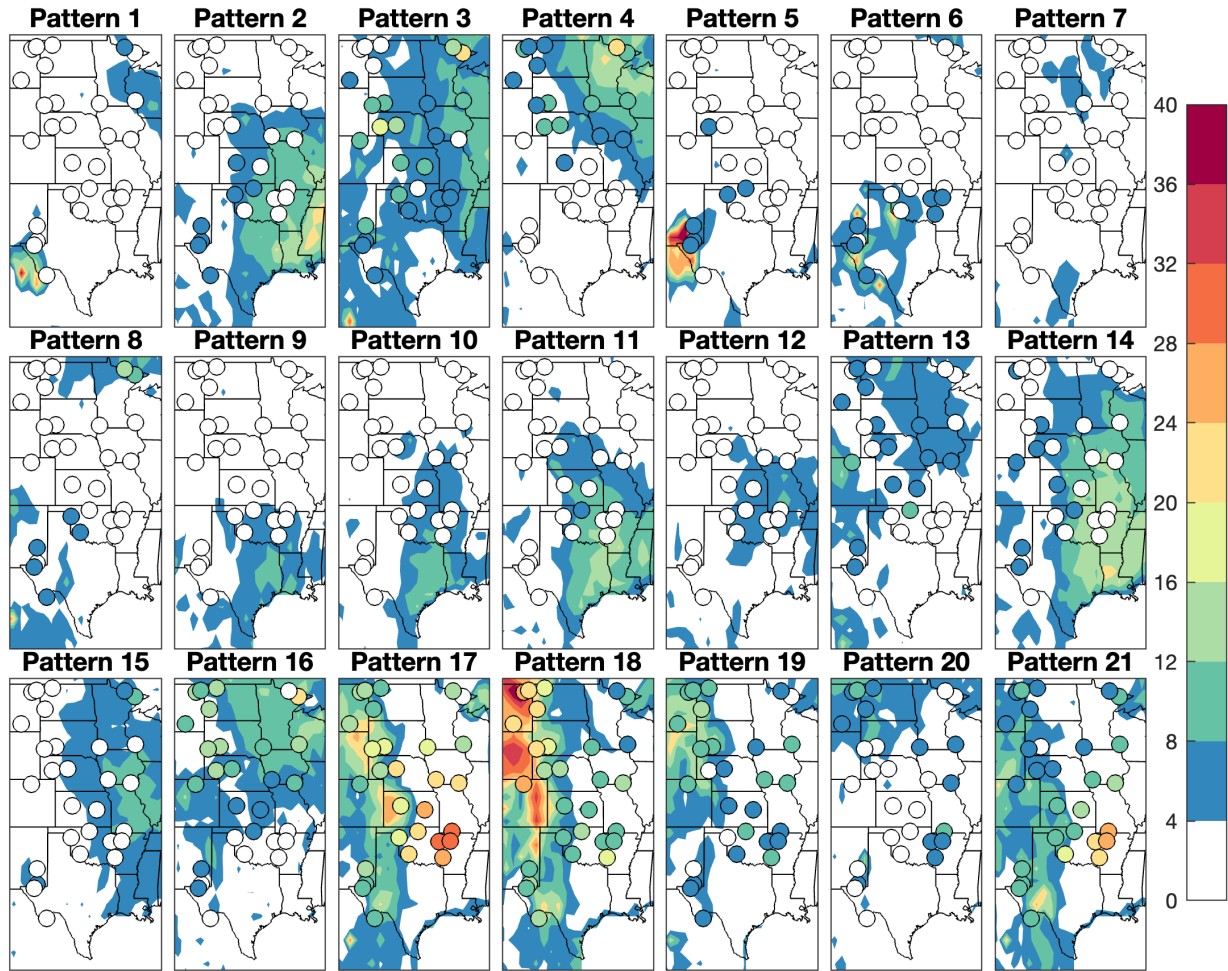

**Figure 4.** Percentage of each location's total VIIRS dust counts (background colors) and IMPROVE days that exceed $1 \mu g/m^3$ (points) that occur during each weather pattern. For both VIIRS and IMPROVE data at a particular location, the sum of the panels is 100%.

Patterns 14-16 each contribute dust to the region with Pattern 16 standing as the second most important contributor. Each of these patterns feature a surface anti-cyclone over the Great Plains at the leading edge of an upper-level ridge. In Patterns 14 and 16, the two anti-cyclonic patterns that produce more dust in the region, the surface high is to the south of the region (southern Missouri and Louisiana/Texas respectively), creating strong southwesterly to westerly winds across the eastern Dakotas and Minnesota. Pattern 15 has the surface high further to the north (centered on Iowa), leading to weaker winds and less dust in the region. IMPROVE also shows significant dust concentrations in the northeast during summer Patterns 17, 18, and 19, something not seen in the VIIRS data. As with VIIRS missing dust in Oklahoma during summer, we consider dust produced by late afternoon convection to be the most likely explanation.

### 3.1.4   The High Plains and Missouri Plateau

The High Plains, in the lee of the Rocky Mountains and at substantial altitude, stretch from the Llano Estacado of the Texas Panhandle and New Mexico northward through western Kansas and eastern Colorado to the Missouri Plateau region of the western Dakotas and eastern Montana and Wyoming. Both datasets agree that dust in this region is primarily a summertime phenomenon and uplift can be initiated by both the strong southerly winds that predominate the season and gust fronts created by local thunderstorms. The E17 classification has five summer weather patterns – Patterns 17-21. All the patterns feature warm surface temperatures, weak pressure gradients, southerly low-level flow, and zonal or anti-cyclonic flow at 500 mb. The slight differences in their meteorology lead to shifts in which parts of the Plains experience precipitation (Figure 1). All five patterns lead to dust occurrence in the western plains, but Patterns 17, 18 and 19 have particularly high dust frequencies. These three patterns all feature more anti-cyclonic 500 mb flow and the hotter surface temperatures (Figure A1), and thus more suppressed convection and dryer soils in the High Plains than the other summer patterns. Patterns 17 and 19 have a strong low-level jet, but Pattern 18 has a relatively weak one, though all have southerly wind speeds which peak in the northern rather than southern plains, helping to uplift dust in the region.

Patterns 3 and 16 both show significant dust in the High Plains in the IMPROVE data but not the VIIRS data. Pattern 16 has a high-pressure system in the eastern plains creating dry weather and modest southerly winds in the High Plains. The VIIRS observations do show dust detections in the southern parts of the High Plains, albeit not at statistically significant levels. More difficult to understand is the discrepancy in observations of dust during Pattern 3. Pattern 3 features a surface low along the Colorado - Kansas border that leads to weak winds in the High Plains and small amounts of rainfall on the Missouri Plateau. As might be expected from such meteorology, VIIRS sees almost no dust in the High Plains during this pattern. IMPROVE, however, reports significant concentrations of dust from Kansas to Montana. One possible explanation for the difference is cloud cover obscuring the view of VIIRS, as Pattern 3 has high cloud cover for this area ranging from 30-60% (not shown). Case studies would be valuable to better understanding this feature.

## 3.2   Primary weather patterns

As Figure 4 shows, one advantage of identifying dust-related weather patterns for different regions is the ability to identify the relative importance of different weather patterns for dust occurrence in a particular region. As described in Section 2, the 21 weather patterns of E17 can be grouped into five categories of weather: southerlies and warm sectors (Patterns 1-4), cold fronts (Patterns 5-8), northerlies (Patterns 9-13), anticyclones (Patterns 14-16), and summer (Patterns 17-21). Summing the information in Figure 4 for each of these five categories of weather provides a summary view of the kinds of weather important to dust in the Great Plains. Figure 5 shows the percentage of VIIRS dust retrievals that occur during each of these five categories.

Figure 5 makes clear the regional variation in dust meteorology across the Great Plains. Each of the five categories of weather dominates within a region according to VIIRS. All along the High Plains from West Texas to the Canadian border, summertime meteorology dominates, being responsible for a majority of dust from Colorado northward. Cold fronts dominate

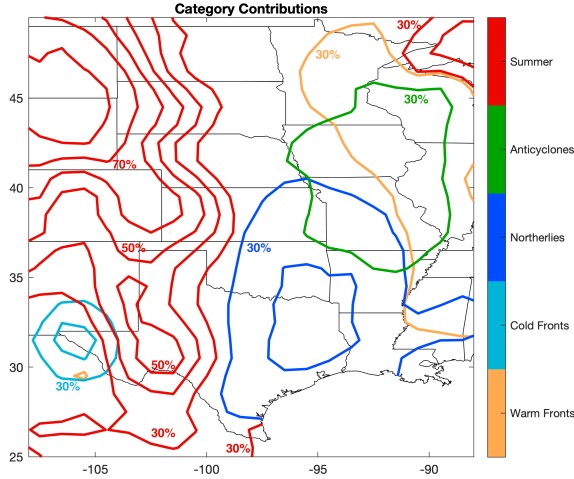

**Figure 5.** Contributions (% of total dust retrievals in each gridcell) for each of the five categories of weather. The lowest contour for each category is 30% with contour intervals of 10%. For a given location, the sum of all five categories contributions is 100%. Patterns within each category are as follows: Southerlies (Patterns 1-4); Cold Fronts (Patterns 5-8); Northerlies (Patterns 9-13); Anticyclones (Patterns 14-16); Summer (Patterns 17-21)

the dust weather of the El Paso region and southern New Mexico. Cold northerlies bring dust to the southeastern plains from Kansas and Missouri to the Gulf Coast, and account for a majority of dust retrievals over eastern Texas. High pressure anticyclones and southerlies are the most important patterns for the northeastern plains of Missouri, Iowa, and Minnesota, with the regions of importance being further north for southerlies and further south for anticyclones. These results largely agree with the IMPROVE results presented in the previous section, with the exception of Oklahoma and the southeastern plains. As noted above, IMPROVE sees summer patterns playing a dominant role in that region and northerlies producing very little dust.

Figure 6 shows the distribution of VIIRS-observed dust contributions from each weather pattern for a selection of cities around the Great Plains. The figure provides greater detail on the results of Figure 5, e.g. summer patterns (red) being particularly important to High Plains cities like Cheyenne and Rapid City, southerlies (orange) bringing dust to Minneapolis, and northerlies (blue) bringing dust to Oklahoma City. Interestingly, it also shows that dust weather is sensitive to the details of the meteorology, as there is substantial within-category variability. For example, southerlies and warm sectors are all contributors of dust to Minneapolis, but it is Pattern 3 and Pattern 4 that are most important, and four northerly patterns contribute substantial dust to Oklahoma City, but the fifth does not. In some cases, it is clear that shifts in the location or direction of strong winds, e.g. cold front Pattern 5 bringing far more dust to El Paso than cold front Pattern 6, but in others it is less clear. Pattern 8, a cold front over west Texas, has a very similar flow pattern to Pattern 5, but does not produce nearly as much dust. Many such examples exist and further investigation of these details would provide value to local-scale understanding of dust weather.

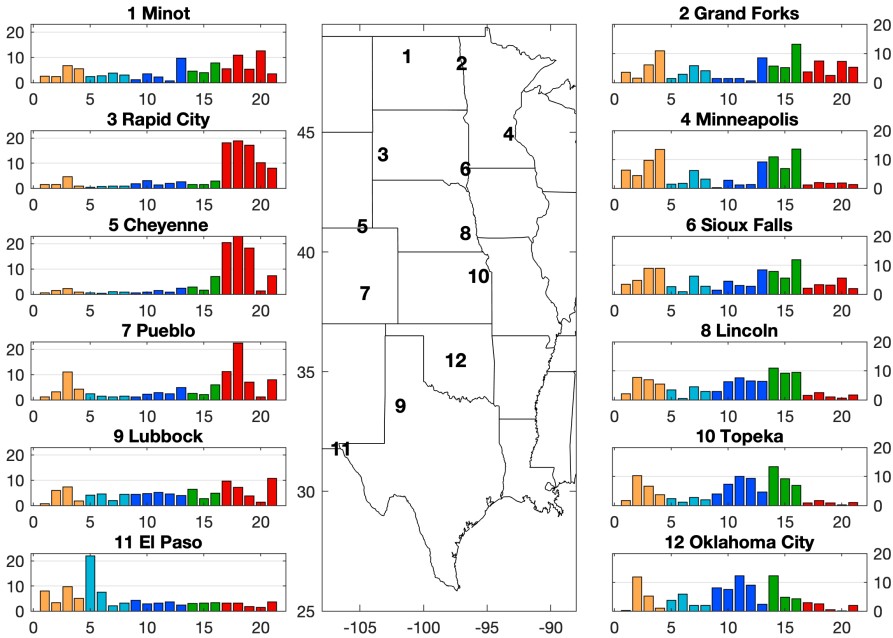

**Figure 6.** Contributions (% of total dust retrievals in each grid cell) for selected cities within the Great Plains from each of the 21 weather patterns. For given city, the sum of the 21 bars is 100%. Weather patterns are grouped and color-coded according to category of weather, as with Figure 5. City locations are indicated by numbers on the map.

## 4 Summary and Discussion

In this study we used a weather pattern classification system for the Great Plains as a basis for compositing Suomi-VIIRS and IMPROVE dust observations in order to determine the meteorology most important for dust occurrence across the region. In previously well-studied regions such as El Paso and the southern high plains of New Mexico and the Texas Panhandle, our findings are in agreement with previous studies, showing that cold fronts extending from low pressure systems leading deep upper-level troughs are the primary source of dust to the region (Novlan et al., 2007; Rivera et al., 2009). In less-studied

portions of the Great Plains our findings are novel. We find that southerly winds and warm sectors are the most important source of dust to the northeastern plains of Iowa, Minnesota, and the eastern Dakotas, while summertime convection is the dominant source of dust in the northwestern high plains from western Kansas and Colorado to Montana and the western Dakotas. In the southeastern plains the two datasets disagree: VIIRS see little dust that primarily occurs during post-frontal northerlies while IMPROVE sees substantial dust concentrations during most summer weather patterns.

The limitations associated with the dust observations add caveats to this study. Satellites undercount dust occurrence due to time of overpass, obscuration by cloud cover, and lack of detection when mixed with other aerosols, thus the VIIRS results in this study undercount dust presence as well. Studies have shown that dust events in the western High Plains generated by

convective outflows occur most frequently during the summer season and late in the day (Novlan et al., 2007; Kelley and Ardon-Dryer, 2021). As a result, VIIRS likely undercounts dust during summer pattens (17-21) more than others, and these may play a more important role than shown above. Nonetheless, the same studies show that synoptically-driven dust events comprise the majority of dust events, so we believe the broad conclusions of the study remain valid. IMPROVE stations do not suffer from these same biases, but come with their own limitations of spatial distribution and only observing every third day. In addition, IMPROVE's 24-hour collection period likely plays an important role in understanding the differences between the two datasets, especially for convectively driven dust during summer. Late afternoon dust events are captured by IMPROVE, so gust fronts or convective systems propagating across the plains over the course of several hours can be captured by many stations from west to east. Together, these effects likely explain the IMPROVE results showing more summer dust in Oklahoma and greater east-west coherence across the northern plains. Further detail into the importance of convective dust events could best be addressed through station data with higher temporal frequency. The weather classification is based on ERA5 reanalysis fields that are available at hourly resolution, so weather pattern can be categorized up to 24 times daily if station data provide dust observations to match.

This study focused on the meteorology that drives dust occurrence in the Great Plains, but the surface properties, or erodibility of the land, are also crucial to determining dust emission. The two are connected via precipitation and relative humidity, but as the land acts as an integrator of weather events, the important properties of soil moisture and vegetation cover vary much slower than atmospheric properties (Evans et al., 2016; Arcusa et al., 2020). The result is that the same weather pattern may produce different amounts of dust depending on the condition of the land surface beneath it. Analysis of why a particular weather pattern sometimes produces dust and sometimes does not with regard to observations of soil moisture, vegetation cover, and snow cover would likely help understand both the within-pattern variability of dust and quantify the importance of land surface properties seasonally and spatially across the Great Plains.

Previous studies have identified trends in the occurrence of dust in the western US (Achakulwisut et al., 2017; Aryal and Evans, 2022) on decadal time scales. Potentially, such trends could be explained in terms of trends in the frequency of important dust-producing weather patterns, however, the short time period analyzed here is a limitation. Only Pattern 5, increasing at 1.1 days/year, has a statistically significant trend (95% confidence) over the period 2012-2021. As this pattern is responsible for a large portion of the dust in the El Paso region of Texas, this trend in pattern frequency may explain the observed increase in springtime dust observed at nearby IMPROVE sites (Achakulwisut et al., 2017; Aryal and Evans, 2022). The frequency of occurrence of weather patterns also has substantial year-to-year variability, and with longer records may explain the interannual variability of dust occurrence in the Great Plains.

Many further analyses are possible using this weather classification as a basis for compositing observations. Many local and regional studies of dust meteorology manually classify dusty days into categories such as "synoptic" and "convective", sometimes sub-dividing those into further categories to account for the variability of observed weather (Brazel and Nickling, 1986; Novlan et al., 2007; Kelley and Ardon-Dryer, 2021; Hahnenberger and Nicoll, 2012). This classification allows expansion on these classifications by providing ready to use, objectively-determined categories with detailed meteorologies applicable across the Great Plains, including in understudied regions such as the northern and eastern Great Plains. This study focused on

using the classification to understand the meteorological causes and contributions to dust; it could also be used to study other aerosols, air quality, air chemistry, or any other phenomena related to weather variability in the Great Plains. The particular record analyzed here, satellite-observed dust occurrence, is relatively recent, so only a small portion of the ERA5 record of weather pattern is used. The ERA5 reanalysis product extends back to 1940 however, so additional studies of long records of aerosols or atmospheric composition could take advantage of weather pattern analysis to investigate the causes of trends and variability over many decades. In doing so, the findings made here, including the causes of dust events across the northern Great Plains, can be used to understand the episodic and understudied events of this complex dust source.

*Data availability.* Time series of weather pattern is archivevd at https://ubir.buffalo.edu/xmlui/handle/10477/85986. ERA5 reanalysis, Suomi-VIIRS satellite data, and IMPROVE observations are available for download from ECMWF, NASA, and Colorado State University respectively.

## Appendix A

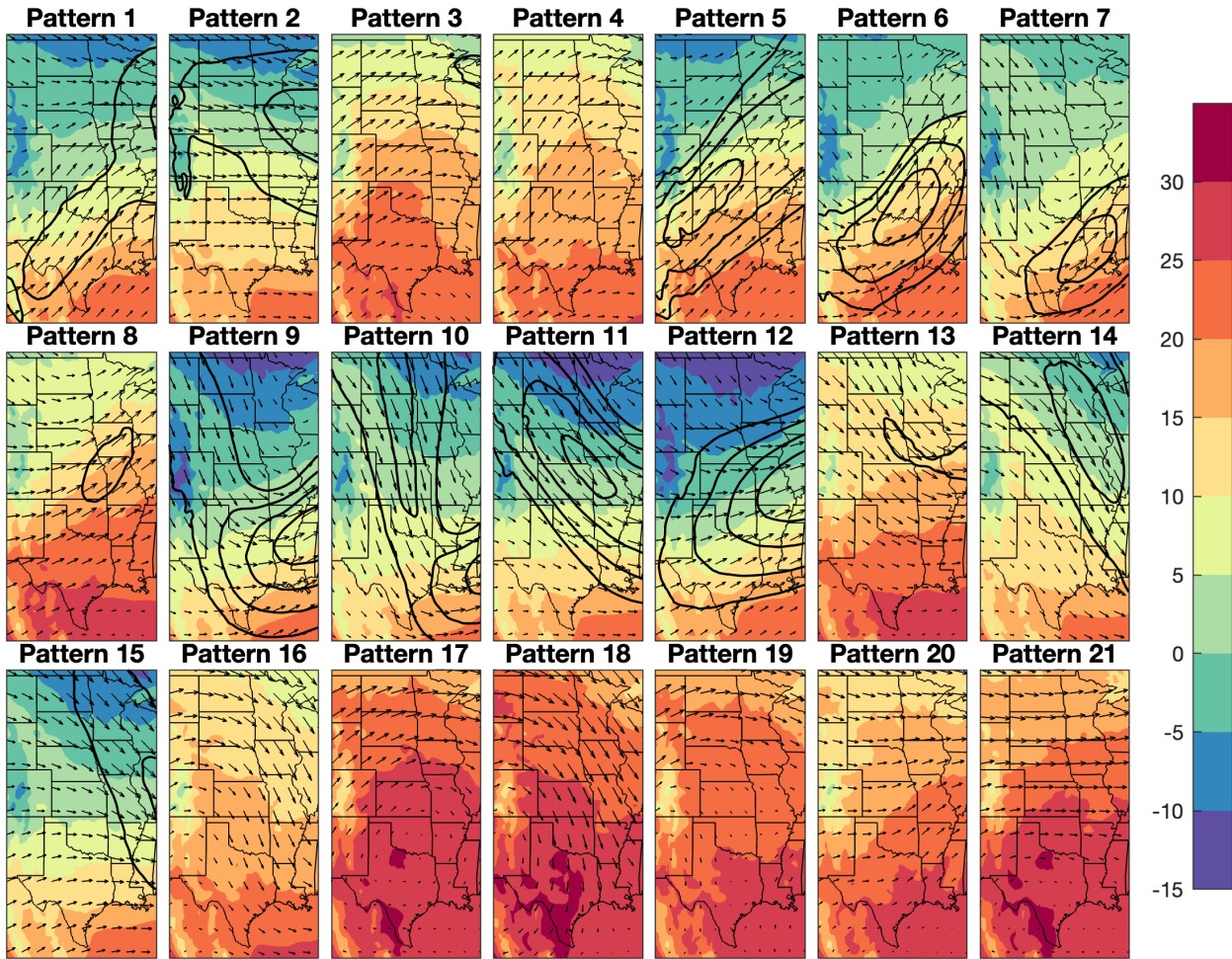

**Figure A1.** Composite 2012-2021 ERA5 properties for each of the 21 weather patterns. Underlying color shows 2m temperature (°C) and arrows show 500mb wind speed and direction, with longest arrows representing 35 m/s. Thick black contours show 500mb wind speed in increments of 4 m/s, beginning at 20 m/s, in order to highlight jetstream location.

*Author contributions.* The author confirms sole responsibility for the following: study conception and design, data collection, analysis and interpretation of results, and manuscript preparation

*Competing interests.* The author declares that they have no conflict of interest.

*Acknowledgements.* This work was supported by the RENEW Institute of the University at Buffalo. We thank the many scientists responsible for developing and making available the ECWMF reanalysis products, the VIIRS satellite products, and the IMPROVE observations used in this study. We thank Dr. Kristin Poinar for helpful comments on an early draft of the manuscript.

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
