# Peer review of "Dust-producing weather patterns of the North American Great Plains"

_EGUsphere, 2024_

## Author Response (AR1)

Response to reviewers

Reviewer 1

We thank Reviewer #1 for their supportive review and close read of the manuscript.

- Line 11: should be of or from, not both.
  - Corrected
- Line 36-37: Something odd happening with the references repeating.
  - Fixed
- Line 93: Spelling - 'wins' should be 'winds'
  - Corrected
- Line 110-111: LT or UTC?
  - Added text specifying local time
- Figure 4: It may be useful to include in the caption the patterns associated with each of the categories.
  - Agreed. Pattern numbers added.
- Line 173: 'southerlies and southerlies'
  - Corrected to "southerlies and warm sectors"
- Figure 5: There appears to be a rogue line on the map extending from the bottom of Texas.
  - Fixed
- Line 214: 'Only Pattern 5, increasing at 1.1 days/year, has a statistically significant trend (95% confidence) over the period 2012-2021'. Was this discussed in the paper? This feels like it has come out of nowhere in the conclusions with no reference to looking at trends previously.
  - Agreed this is out of place. In response to Reviewer 2 there is now a new piece of analysis using many IMPROVE stations throughout the plains. This line is removed from the trends discussion and integrated into discussion of the new IMPROVE results in the context of previous studies using IMPROVE data.

Reviewer 2

We thank Dr. Adebiyi and Mr. Ebiendele for their thoughtful review of the manuscript. We also wish to commend the mentorship of Dr. Adebiyi for his inclusion of a graduate student in the review process. All comments are responded to individually below.

We take note of the overarching concern in many of the comments for the quality and potential biases from using exclusively the VIIRS aerosol classification product. While we believe that this is a high-quality data product, the we have been careful in our interpretation of the results, and provided sufficient caveats where appropriate (as noted by Reviewer #1), we do recognize the larger point that an additional data source would substantially strengthen the conclusions of the study. As such we have opted to add station data from 29 members of the IMPROVE network, spread across the plains, to our analysis. This data directly speaks to the questions raised below of satellite biases (clouds, overpass time, retrieval). The IMPROVE results broadly confirm those of VIIRS, with some interesting differences. We discuss those differences in the context of possible observation biases. We believe this added analysis and discussion addresses the major concerns in the review.

We include here an example figure of the IMPROVE data, using average concentration of soil (ug/m3) as a metric. It shows broad agreement with the VIIRS results that (a) Pattern 5 enhances dust in the El Paso region, (b) Patterns 3, 4, and 16 enhance dust in the northern plains, and (c) summer patterns 17, 18, and 19 enhance dust in the high plains. It shows differences in the northern plains having less east-west separation than in VIIRS, Pattern 3 enhancing dust in the El Paso region, and Pattern 21 enhancing dust in Oklahoma. These similarities and differences are discussed in terms of the differences between the observations.

- While the classification method was originally defined in a different paper (Evans et al. 2017), the author provided a brief summary that was largely limited in delivering the needed overview necessary for a reader who may not want to go through the other paper. For example, it is not clear in this paper why 21 patterns were identified or the reason for limiting the Great Plains to the area defined. These issues must be addressed here to have a complete paper and not leave readers in a state of confusion.
    - We have edited the description of the classification process to clarify the iterative process, its automated nature, and that this process found 21 patterns. We have also added a sentence describing the purpose of the E17 classification to make clear why it is focused on the Great Plains.
- In addition, the authors identified several uncertainties associated with the observational data used for dust identification. Despite these uncertainties, it is unclear why the author decided to use this dataset over other equally available and potentially better datasets. For example, the author cited Ginoux's paper that used MODIS. With a better dust-retrieving algorithm than VIIRS, the author did not provide a valid reason for their decision or how potentially the results would be different if

this other dataset was used. We suggest that additional analysis should be done with at least one other similar dust observation that will mitigate some of the uncertainties identified with dust retrieval in VIIRS.

- o MODIS does not provide an aerosol classification product similar to one used here and recreating the dust optical depth product from Ginoux, et al. (2012) is beyond the scope of this study. MODIS does provide aerosol optical depth, but this would include the effects of other aerosols present in the region such as smoke and urban pollutants, making it unsuitable for this study. Further, we maintain that the aerosol data provided by VIIRS is high-quality: comparisons of AOD between VIIRS and AERONET are as good as between MODIS and AERONET (Hsu et al. (2019) Figures 15-17 and Table 6). Specific to the Great Plains, VIIRS and the two MODIS instruments perform very similarly (Sayer et al. (2019) Figures 3, 15-17). We have added text in the Dust Observations subsection of Methods stating that we choose VIIRS data for the aerosol classification product and commenting on the quality of VIIRS aerosol data. More importantly, we have added analysis of IMPROVE data to complement the analysis of VIIRS data.

- Also – there is a fair amount of confusion around how the composite analysis for dust was performed. The author mentioned that four times daily ERA-interim datasets were used for the weather pattern (Line 75), but daily observation was used for the VIIRS dust (Line 117). How does the author reconcile that difference? Similar to the comment above, this suggests that a better temporal dust retrieval, such as dust RGB from GEOS, maybe a better alternative than the VIIRS used in this analysis.
  - o This was stated in the final sentence of the Classification Process subsection, but we have rewritten it to make the process clearer and more explicit.

Other Comments:

- Line 11: remove "of"
  - o Corrected
- Line 83: Do you mean A1 instead of S1? Also, why is Fig. A1 in the main manuscript?
  - o Corrected to A1. This manuscript was prepared using the Copernicus Publications LaTeX Package, provided by ACP, which places Appendix A where it appears.
- Are Fig. 1 and A1 showing for 2012-2021 that are used for dust classification, or are they showing for 1996-2010 used in the other paper? This should be identified in the caption and stated clearly in the text. Given that 1996-2010 is entirely irrelevant to

the results of the dust pattern shown in Fig. 3, 4, and 5, I hope Fig. 1 and A1 are showing for 2012-2021. Otherwise, the figure should be changed and results re-interpreted if the pattern is different.

- o The meteorology shown in Figures 1 and A1 were for the combined period, but we have remade the figures to only show the dust observation period. Neither the patterns nor the values vary in any notable way as they are, by definition, required to closely match the patterns identified in E17.

- Line 76-80: The authors used a "k-medians classification algorithm to identify and define commonly occurring weather patterns for the region." There are many other clustering methods (e.g., hierarchical, K-means); why do the authors consider the K-medians algorithm? I recommend that the authors provide a more detailed justification for their decision.

- o The methodology of Evans, et al. (2017) follows that of Evans, et al. (2012), Marchand, et al. (2009), and Marchand, et al. (2006). We have updated the description in Methods to reflect that this methodology has been developed over several publications and successfully used in the analysis of both cloud and radiation observations in the Great Plains (Muhlbauer, et al. (2014), Zhao, et al. (2019)). We believe that a discussion of the relative merits of different clustering algorithms is outside the scope of this study.

- Lines 111-115: The authors stated, "We do not capture dust that initiates after the overpass or occurs beneath clouds." Possible bias could result from undercounting, especially during seasons with higher cloud cover. Could the authors provide details on the potential significance of undercounting in regions or seasons with high cloud cover?

- o This is addressed by the introduction of IMPROVE data. We comment on the possibly influence of cloud cover on VIIRS observations when comparing the two. Cloud cover in the Great Plains is more common in the northeast than southwest and in winter rather than summer.

- Lines 145-146: The authors' findings indicate that "Pattern 4" is the primary source of dust in the region, as illustrated in Figure 3. Could the authors explain how this aligns or differs from previous studies over the northeastern plains?

- o The northeastern plains are a heavily understudied region for dust events. We are unaware of any previous studies identifying weather patterns associated with dust events in the region. As such, we refer to our results for the region as novel.

- Line 220-222: "The summer weather patterns (Patterns 17-21) likely undercount dust more than others. Since convective summer dust events are likely frequent and short-lived, this could be a significant constraint. I suggest the authors discuss

how to address this bias. The authors could discuss possible alternative satellite products or station-based observations to address this.

- o   Again, the added analysis of IMPROVE data addresses this question.

---

## Referee Report (RR1)

**Referee Comment on Revised Manuscript**

I appreciate the author careful attention to all the comments. The revised manuscript has improved clarity on the potential bias from the use of the VIIRS aerosol classification product through the inclusion of the IMPROVE station data, providing more robust confidence in the results.

The author addressed all major and minor concerns; the revised manuscript now provides a clearer explanation on the 21 identified patterns and validated their reliance on the VIIRS product to reduce uncertainties from satellite-only bias. The inclusion of spatial discrepancies between the VIIRS aerosol classification product and IMPROVE observational data, especially in the southern plains of Oklahoma and Kansas, is also helpful for the readers.

The revised manuscript has improved clarity and now provides a more robust and well-validated assessment of how different identified weather patterns drive dust across the North American Great Plains. I believe that the revised manuscript has addressed the main concerns and that the manuscript is notably stronger for it.

I recommend acceptance, contingent on any final minor editorial checks.